# Characterization of Influenza D Virus in Danish Calves

**DOI:** 10.3390/v14020423

**Published:** 2022-02-18

**Authors:** Nicole B. Goecke, Yuan Liang, Nina D. Otten, Charlotte K. Hjulsager, Lars E. Larsen

**Affiliations:** 1Department of Veterinary and Animal Sciences, University of Copenhagen, 1870 Frederiksberg, Denmark; yuan.liang@sund.ku.dk (Y.L.); nio@sund.ku.dk (N.D.O.); lael@sund.ku.dk (L.E.L.); 2Virus and Microbiological Special Diagnostics, Statens Serum Institut, 2300 Copenhagen, Denmark; ckhj@ssi.dk

**Keywords:** influenza D virus, bovine, calves, PCR, phylogeny, Denmark, sequencing

## Abstract

Influenza D virus (IDV) was first described in 2011 and has been found to mainly circulate among cattle and swine populations worldwide. Nasal swab samples were collected from 100 Danish calf herds (83 dairy and 17 veal herds) from 2018–2020. Influenza D virus was detected in 12 of the herds. Samples with the lowest cycle quantification value were selected for full genome sequencing. A hemagglutinin-esterase fusion (HEF) gene sequence from a Danish IDV collected in 2015 was also included in this study. Phylogenetic analysis showed that viruses from seven of the IDV-positive herds belonged to the D/OK lineage and clustered together in the HEF tree with the IDV collected in 2015. Viruses from the four other herds belonged to the D/660 lineage, where three of the viruses clustered closely together, while the fourth virus was more phylogenetically distant in all gene segments. The high level of genetic similarity between viruses from two different herds involved in calf trading suggests that transmission occurred through the movement of calves. This study is, to our knowledge, the first to describe the characterization of IDV in calves in Denmark.

## 1. Introduction

Influenza D virus (IDV) is a segmented negative-sense single-stranded RNA virus that belongs to the family Orthomyxoviridae, which also contain the species influenza A virus (IAV), B virus (IBV), and C virus (ICV). The genomes of ICV and IDV consist of seven gene segments, while the genomes of IAV and IBV consist of eight gene segments. The genome of IDV encodes nine proteins, including glycoprotein hemagglutinin-esterase fusion (HEF), which combines the functions of receptor recognition and binding, receptor destroying, and membrane fusion. IDV is genetically most closely related to ICV, but they share only around 50% amino acid identity for HEF and 70% for polymerase basic protein 1 (PB1), which is the most conserved segment. Furthermore, no cross reactivity between human ICV antisera and IDV has been seen [1,2,3]. 

IDV was isolated for the first time in 2011 from pigs exhibiting an influenza-like illness in Oklahoma in the USA [2]. However, serological evidence has subsequently indicated the circulation of IDV among cattle in Nebraska since 2003 [4]. Besides cattle and pigs, horses, sheep, goats, camelids, and feral swine have been found to be IDV-seropositive, which suggests a broad host-range of the virus [5,6,7,8]. Despite the broad host-range, it is believed that bovines are the natural reservoir of IDVs, due to the frequent isolation of these viruses from cattle and the finding of high seroprevalences in bovine herds [5,9]. Since the first discovery, IDVs have been detected in cattle in regions of the USA, Mexico, Canada, Asia (China, Japan), and Europe (France, Ireland, Italy, Luxembourg, the United Kingdom) [4,10,11,12,13,14,15,16,17,18]. The first European report of IDV in cattle was in 2012 in France [14].

Metagenomic studies have shown that IDVs can be associated with bovine respiratory disease (BRD), which is a multifactorial disease associated with different pathogens, stress conditions, environmental factors, and the health and immunological status of the animal [10,19]. Bacterial pathogens included in BRD are *Mannheimia haemolytica*, *Pasteurella multocida*, *Mycoplasma bovis*, and *Histophilus somni*, while viral agents known to participate in BRD are bovine adenovirus, bovine coronavirus, bovine herpesvirus 1, bovine parainfluenza virus type 3, bovine respiratory syncytial virus, and bovine viral diarrhea virus [20,21,22]. Only a few of these pathogens can successfully cause severe disease alone. The role of IDV in BRD in the field is still not clear, and further studies are needed to clarify this [23]. Experimental and field studies have shown that IDVs can cause mild to moderate respiratory signs in cattle, and that viruses are able to replicate in both the upper and lower bovine respiratory tract; however, they seem to have a preference for the upper tract [9,24,25,26].

The zoonotic potential of IDV is still unclear, and so far, there are no indications that IDV can cause disease in humans nor be transmitted among humans. However, studies have shown that IDV is able to transmit and replicate by direct contact in guinea pigs and ferrets, which are both established models for human influenza virus infection [2,27]. In addition, IDV-specific antibodies have been detected in humans with occupational exposure to cattle [28], and in the general population in Italy [29]. Another study demonstrated that IDV replicates efficiently in well-differentiated human airway epithelial cells, which act as an in vitro respiratory epithelium model for humans [30]. The mentioned studies support the hypothesis that IDV has zoonotic potential.

So far, IDVs have been phylogenetically classified into two major lineages, D/OK and D/660, in Europe and North America, while different lineages (D/Yama2016, D/Yama2019) have been found in Japan [31,32,33]. D/OK is thus far the most frequent reported lineage circulating in Europe, and until 2018, D/660 was unreported in this geographic area. However, an Italian IDV surveillance study performed in 2018–2019 found samples belonging to both the D/660 and D/OK lineages, showing that viruses from both lineages circulated in Italian cattle. The earliest European D/660 strain was detected in Italy in March 2018 from cattle imported from France. Since 2019, more D/660 strains than D/OK strains have been identified in Italy suggesting a shift in genotypes [33]. IDV strains with a reassortant genetic pattern, containing gene segments from both D/OK and D/660, have been reported in Italy and the USA [32,33]. Besides the circulation of the D/OK and D/660 lineages in Europe, a genetically divergent lineage containing two strains (D/bovine/France/2986/2012 and D/bovine/Ireland/007780/2014) has been described [14,16]. However, further sampling and sequencing is needed in order to be able to assess whether this represents the circulation of a new lineage in Europe. Furthermore, a different phylogenetic group of IDV has recently been identified in California and named D/CA2019 [34].

While IDVs have been shown to only cause mild or moderate respiratory disease in cattle, the continued circulation of IDV in European cattle and swine herds, combined with the potential for zoonotic transmission, prompts further elucidation of IDVs and their evolution. IDV status remains uninvestigated in many countries. Therefore, the aim of the present study was to genetically characterize IDVs in Danish calves to obtain more knowledge on the diversity of these viruses in Denmark.

## 2. Materials and Methods

### 2.1. Samples, Clinical Examination, and Herd Descriptions

Nasal swab samples were collected as part of a Danish research project where 100 Danish, intensive, commercial herds (83 dairy and 17 veal herds) were sampled. In Denmark, a veal herd is a rosé veal calf producing unit, where mainly bull calves are purchased from Danish dairy herds and raised for meat production [35]. Sampling was done in the period from September to April in 2018–2019 and 2019–2020, either once in each herd or at multiple time points. The samples were collected from three age groups in the dairy herds (0–10 days, 2–4 weeks, and 2–4 months), and two age groups in the veal herds (2 weeks after arrival (in general, 3–5 weeks of age) and at 3 months of age). The nasal swab samples were collected by inserting a sterile cotton swab approximately 8–10 cm into one nostril and turning the swab around for a few seconds, and immediately after, the swab was placed and stored in 1.5 mL phosphate-buffered saline (PBS). No prior cleaning of the nostril was performed. Further information about the handling and storage of the samples is described elsewhere [36].

A previous study investigated the occurrence of 11 respiratory and enteric pathogens in the 100 Danish calf herds at the first sampling time using a high-throughput real-time PCR platform, BioMark HD (Fluidigm, South San Francisco, CA, USA), and the 192.24 (samples.assays) dynamic array integrated fluidic circuit nanofluidic chip (Fluidigm) [36]. The herds that were sampled at multiple time points were also tested on the high-throughput real-time PCR (data not shown). Collectively, in the high-throughput real-time PCR analyses, samples (either individual samples and/or pools) from 1 dairy and 11 veal herds tested positive for IDV (Appendix A). In the present study, samples from some of the IDV-positive pools were subsequently tested individually by single-plex real-time reverse transcription-PCR (real-time RT-PCR) (Appendix A). The samples with the lowest cycle quantification (Ct) value, as determined by real-time RT-PCR, was selected for further whole-genome analysis, meaning that each of the 12 herds were represented by one sample, with the exception of Herd ID 5, where six samples from three different time points were included (Table 1). Furthermore, a bronchoalveolar lavage sample collected in 2015 from a calf in a veal herd was also included. Clinical observations or any other information were not available for this animal. Altogether, IDV sequences from 18 samples, representing 13 herds, were analyzed in this study. An overview of the IDV-positive pools and individual samples can be found in Appendix A.

At each sampling, clinical signs (coughing, rectal temperature, nasal and ocular discharge) were recorded for each sampled calf and scored based on a scoring system that was adapted from two existing scoring systems [37,38]. For each sign, clinical scores were rated depending on the severity, and a score of zero was given if the sign was not present. For coughing, the lowest score (zero) was given if no coughs were observed and the highest score (three) if repeated unprovoked coughs were observed. For nasal and ocular discharge, serous discharge equaled a score of one, while mucopurulent discharge equaled a score of two. Body temperature was measured rectally and categorized as either normal at temperatures below 39.3, or as febrile when exceeding this cut-off.

### 2.2. RNA Extraction

For RNA extraction, 200 µL nasal swab sample suspension was transferred to a tube containing 400 µL RLT-buffer (QIAGEN, Copenhagen, Denmark) containing 1% 2-Mercaptoethanol (Merck, Darmstadt, Germany). The RNA was subsequently extracted from the sample using an RNeasy mini kit (QIAGEN) on the extraction robot QIAcube Connect (QIAGEN) using the large sample protocol, version 2, and elution volume 60 µL, according to instructions from the supplier.

### 2.3. Real-Time RT-PCR

Real-time RT-PCR targeting the PB1 gene of IDV was used with the following primers: D/OK forward 5′-GCT GTT TGC AAG TTG ATG GG-3′, D/OK reverse 5′-TGA AAG CAG GTA ACT CCA AGG-3′, D/OK probe 5′-[FAM]-TTC AGG CAA GCA CCC GTA GGA TT-[BHQ1]-3′ [2], and with AgPath-IDTM One-Step RT-PCR Reagents kits (Applied Biosystems^TM^, Foster City, CA, USA). The real-time RT-PCR was performed in a final volume of 25 µL with 5 µL extracted RNA template. A total of 12.5 µL RT-PCR buffer (2X) was mixed with 1.50 µL of each primer (10 µM), 0.75 µL probe (10 µM), 1 µL RT-PCR enzyme mix (25X), and 2.75 µL nuclease-free water. PCR reactions were run on Rotor-Gene Q (QIAGEN) using the thermal cycling conditions: 48 °C, 30 min (min); 94 °C, 10 min; 45 cycles x (94 °C, 15 s (s); 60 °C, 45 s). All reactions were run in duplicates and were considered positive if they had a Ct value < 36.

### 2.4. Sanger Sequencing

For the amplification of HEF PCR products of the IDV from 2015 for Sanger sequencing, previously described primers were used [14], along with the SuperScript^TM^ III One-Step RT-PCR System with Platinum^TM^ Taq High Fidelity DNA Polymerase kit (Invitrogen^TM^, Thermo Fischer Scientific, Roskilde, Denmark). The RT-PCR was performed in a final volume of 40 µL with 5 µL RNA. A total of 20 µL reaction mix (2X) was mixed with 0.20 µL of each primer (100 µM), 0.80 µL enzyme mix, and 13.7 µL nuclease-free water. The RT-PCR was run on a T3 Thermocycler (Biometra, Fredensborg, Denmark) with the following thermal cycling conditions: 50 °C, 30 min; 94 °C, 2 min; 5 cycles x (94 °C, 10 s; 60 °C, 30 s; 68 °C, 45 s); 35 cycles x (94 °C, 10 s; 57 °C, 30 s; 68 °C, 45 s); 5 cycles x (94 °C, 10 s; 54 °C, 30 s; 68 °C, 45 s); 68 °C, 5 min. The PCR products were visualized on 0.8% E-gel^TM^ general purpose agarose gels (Invitrogen^TM^). The PCR products were subsequently purified with the High Pure PCR Product Purification Kit (Roche, Hørsholm, Denmark). The purified products were then Sanger sequenced with the aforementioned PCR primers at LGC Genomics (Berlin, Germany).

### 2.5. Whole-Genome Sequencing

For whole-genome sequencing (WGS), a modified version of a previously described one-tube PCR protocol was performed [39]. First, previously described primers were used [1] along with the SuperScript^TM^ III One-Step RT-PCR System with Platinum^TM^ Taq High Fidelity DNA Polymerase kit (Invitrogen^TM^). The RT-PCR was performed in a final volume of 40 µL with 4 µL being from the extracted RNA. A total of 20 µL reaction mix (2X) was mixed with 0.16 µL of each primer (50 µM), 0.80 µL enzyme mix, and 14.9 µL nuclease-free water. The PCR was run on a ProFlex^TM^ PCR System (Applied Biosystems) under the following thermal cycling conditions: 42 °C, 60 min; 94 °C, 2 min; 5 cycles x (94 °C, 30 s; 45 °C, 30 s; 68 °C, 3 min); 31 cycles x (94 °C, 30 s; 57 °C, 20 s; 68 °C, 3 min); 68 °C, 7 min. The amplified mixtures were visualized on 0.8% E-gel^TM^ (Invitrogen^TM^) and purified with the High Pure PCR Product Purification Kit (Roche). Libraries were built with the Nextera XT DNA Library Preparation Kit and sequenced on the Illumina MiSeq system (Illumina, Copenhagen, Denmark) with MiSeq Reagent Kit v2, 500 cycles, according to instructions from the supplier.

### 2.6. Consensus Sequence Generation

Data from the Sanger sequencing was imported into the CLC Main Workbench (version 20.0.3) (QIAGEN), where the sequences were assembled against a reference sequence of the HEF gene (D/bovine/Italy/46484/2015). The WGS fastq data were imported as paired-end to the CLC Genomics Workbench (version 21.0.3) (QIAGEN) where reads were trimmed using default settings. The reads were subsequently mapped against reference sequences for each gene segment (D/swine/Italy/199724-3/2015). Consensus sequences were then extracted for each gene segment. Proofreading of the sequences was performed manually. The sequences of this study are available with the accession numbers OM468185–OM468297 (Appendix A).

### 2.7. Maximum Likelihood Phylogeny

Complete IDV sequences available were selected from the GenBank database (NCBI, http://ncbi.nlm.nih.gov (accessed on 3 November 2020)) for phylogenetic analysis, as well as sequences based on high percentage identity by nucleotide BLAST searches. The sequences were aligned with the Danish IDV sequences using the CLC Main Workbench (Version 20.0.3). Model testing was performed in the CLC Main Workbench (Version 20.0.3) (QIAGEN) for each gene segment (PB2, PB1, P3, HEF, NP, P42, NS) alignment to determine the best fitting substitution model. All gene segments, except for NS, were best described by GTR + G + T. NS was best described by a HKY + G + T substitution model. Maximum likelihood phylogeny analysis with 1000 bootstrap replicates was performed in the same program for each gene segment, with the appropriate substitution models selected. Rate variation was included and the analysis was set to estimate gamma distribution parameters. The remaining settings were kept at default. All trees were rooted on gene segments from C/Bovine/Montana/12/2016. The root was subsequently removed. The resulting trees were visualized in FigTree (version 1.4.4) [40].

### 2.8. Prediction of Glycosylation Sites

To predict the N-linked glycosylation sites of HEF, the bioinformatics tool NetNGly–1.0 (http://www.cbs.dtu.dk/services/NetNGlyc/) (accessed on 10 November 2021) was used.

## 3. Results

### 3.1. Clinical Signs

The clinical registrations and observations for the IDV-positive calves included in the analysis are listed in Table 1. Additionally, some IDV-positive calves were also co-infected with other bovine respiratory pathogens (Appendix A). Three of the calves coughed repeatedly by provocation (score of two) or without provocation (score of three), while no coughing was observed for the rest of the animals (score of zero). Nasal discharge was observed for all of the calves as either serous (score of one) or mucopurulent (score of two) discharge. Ocular discharge was not observed for eight calves (score of zero), while either serous (score of one) or mucopurulent (score of two) discharge was observed for the rest of the animals. Rectal temperature was measured to be between 38.3–40.5 ℃. Clinical observations and information about sex and age were not available for the calf sampled in 2015 (D/bovine/Denmark/47398-3622/2015-11-03).

### 3.2. WGS and Phylogenetic Analysis

WGS was obtained from samples from 11 of the selected herds (Table 1, Herd ID 1–11), while sequencing was unsuccessful for Herd ID 12, likely due to low viral load in the sample. For the IDV-positive sample from 2015 (Table 1, Herd ID 13) only the HEF gene was sequenced using Sanger sequencing.

The gene segments of the Danish IDVs had a nucleotide sequence identity of 94.84–100% and amino acid sequence identity of 93.88–100%. All seven gene segments were individually analyzed by inferring phylogenetic trees using the maximum-likelihood method. All gene segments of Danish IDVs collected in 2019, as well as one virus from 2020 (D/bovine/Denmark/3313408142/2020), clustered together (Figure 1 and Figure 2) and can be phylogenetically classified as belonging to the D/OK lineage (Figure 1). The Danish viruses in this lineage had a nucleotide sequence identity of 98.19–100% and amino acid sequence identity of 98.04–100%. The HEF segment sequenced from the sample from 2015 also clustered within the D/OK lineage. Six of the eleven analyzed Danish viruses from 2019–2020 were from the same herd, but collected at three different time points (Table 1, Herd ID 5). From this herd, all gene segments of the four viruses sampled on 15 January 2019 were almost identical (99.80–100% nucleotide identity and 99.74–100% amino acid identity), whereas they were slightly more different from viruses sampled in the same herd on 20 February 2019 and 24 April 2019 (99.25–100% nucleotide identity and 99.09–100% amino acid identity) (Figure 1 and Figure 2). Notably, the latter two viruses clustered with D/bovine/Denmark/6171103755-1/2019 collected in a dairy herd on 6 February (Herd ID 1), that had sold calves to Herd ID 5, which were collected and transported to the facility by staff members working at Herd ID 5. Geographically, these two herds are also closely located (approximately 22 km apart). One of the viruses collected in 2019 (D/bovine/Denmark/5246407608-2/2019) clustered in another branch and shared the highest degree of similarity in the HEF gene (99.53% nucleotide identity) to the virus collected in 2015 (D/bovine/Denmark/47398-3622/2015). D/bovine/Denmark/5246407608-2/2019 was collected in a herd that was located approximately 50 km away. The remaining D/OK viruses were collected in herds that were not closely located nor were connected by trade. Overall, the Danish D/OK IDVs clustered closest to IDVs collected from bovine and swine in Italy, France, and Northern Ireland, while they were more phylogenetically distant from IDVs collected in the USA and China (Figure 1). There was no evidence of reassortments among the Danish IDV strains since the viruses showed similar clustering in all phylogenetic trees based on the different gene segments (Figure 1 and Figure 2).

All gene segments of the Danish IDVs collected in 2020 clustered together except for D/bovine/Denmark/3313408142-6/2020 and belonged to the D/660 lineage. The nucleotide sequence identity for the Danish viruses in this lineage was 96.32–100% and the amino acid sequence identity was 98.35–100%. D/bovine/Denmark/3727003200-7/2020 was the most phylogenetically distant for all gene segments in comparison with the three remaining IDVs from 2020 (Figure 1 and Figure 2). The Danish D/660 IDVs clustered closest together with IDVs collected from bovines in Italy, while they were more phylogenetically distant from IDVs collected in North America (Figure 1). Geographically, the three herds, where D/bovine/Denmark/5256205576-8/2020 (Herd ID 8), D/bovine/Denmark/5995505714-9/2020 (Herd ID 9), and D/bovine/Denmark/1052304114-10/2020 (Herd ID 10) were collected, are located within a vicinity of 89 km, whereas D/bovine/Denmark/3727003200-7/2020 was collected from a herd (Herd ID 7) situated between 216-268 km from the other herds. Similarly, there was no known trade connections between the Danish herds with D/660 IDV detections.

### 3.3. Prediction of Glycosylation Sites

The prediction of the N-linked glycosylation sites of HEF showed the presence of several glycosylation sites for the D/OK and D/660 lineages. The bioinformatics tool NetNGly predicted the positions 28, 54, 146, 249, 346, 513, and 613 to be glycosylated with different specificities, and the glycosylation sites were present in both lineages (Appendix A). For the Danish D/OK sequences, three of these differed by having one less glycosylation site. In the D/bovine/Denmark/5195604790-5/2019-02-20, Thr 56 was replaced by Asn, which transforms the glycosylation site (position 54–57) from a highly specific site to a non-glycosylation site. In the D/bovine/Denmark/5246407608-2/2019-03-01 and D/bovine/Denmark/47398-3622/2015-11-03, Thr 515 was replaced by Ile which changes the glycosylation site (position 513–516) to a non-glycosylation site. The Danish D/660 sequences showed a similar N-glycosylation site prediction pattern.

## 4. Discussion

IDV has mainly been detected in ruminants, with cattle being considered the primary reservoir. The prevalence of IDV has not been elucidated in animal populations in Denmark. Here, we performed the characterization of IDV detected in a research project where 100 cattle herds across Denmark were tested for the presence of bovine pathogens, including IDV using a high-throughput real-time PCR system [35]. Thus, the samples did not originate from diagnostic submissions. Of the investigated herds, IDV was detected in 12 out of 100 herds. Some of the infected calves displayed minor clinical signs such as coughing, discharge from the eyes and/or nose, and slightly elevated rectal temperatures. Whether these clinical signs were caused by the IDV infection alone is not known, as the animals could also be positive for other pathogens such as bovine coronavirus, *M. bovis*, *Mycoplasma*, *M. haemolytica*, *H. somni, P. multocida*, or *T. pyogenes*. Nevertheless, these signs correspond well with an upper respiratory viral infection and are similar to findings elsewhere [9].

Genome characterization of the Danish IDVs showed that the viruses collected in 2015 and 2019 belonged to the D/OK lineage, while most of the viruses collected in 2020 belonged to the D/660 linage. Until 2018, the D/660 linage was unreported in Europe, and the first European detection was in Italy in March 2018 from cattle imported from France. Subsequently, more D/660 strains than D/OK strains are being found in Italy, indicating an ongoing shift in genotype [33]. Despite the limited number of samples analyzed in the present study, a similar shift in genotype also seem to be taking place in Denmark.

Our phylogenetic analysis revealed that the Danish IDVs detected in the herds from 2019 and one from 2020 are the most closely related to the first Danish IDV detection which was from 2015, suggesting that all D/OK-like viruses in Denmark could originate from a single introduction, but due to a gap in surveillance between 2015 and 2019 this remains speculative. Similarly, based on the similarity of the D/660-like viruses, it is likely that they share a common ancestor, indicating a single incursion followed by genetic drift or multiple incursions of almost identical viruses. Herd ID 5 was sampled at three different time points in 2019, with the viruses found in January clustering together and viruses from February and April being more closely related. Since only a part of the calves from the herd were sampled, we are unable to determine whether the distant viruses from February and April are due to new incursions or whether they had already been circulating in January. Since this is the first screening for IDVs in Denmark, the route of incursion of IDV into Danish calves and how long IDVs have been circulating in Denmark is also unknown. One route of incursion of IDV could be the import of cattle from countries infected with IDV; however, the annual import of living cattle is very limited, with less than 200 imported cattle per year from other European countries (unpublished data). Most of the cattle imported since 2017 originate from Sweden, Germany, and Finland, with some being from France. Further IDV surveillance in other European countries is needed elucidate the circulation and route of introduction of IDVs across borders. Within Denmark, trade with calves is common, where bull calves are often sold from dairy herds to veal herds at an early age. In the present study, the owner of one of the veal herds (Herd ID 5) bought and directly transported calves from a dairy herd (Herd ID 1). Notably, the viruses from these two herds were closely related which suggests inter-herd transmission by infected calves. To our knowledge, there is no connection among the remaining IDV-positive calf herds.

Amino acids in the HEF gene under positive selection have been identified, and it is believed that these might affect the receptor binding cavity relevant for broader cell tropism. One study identified two amino sites to be under positive selection (residues 251 and 289), both of which are located in the globular domain of HEF [41]. The amino acids Ala and Thr were found to be present at residue 251, while in the Danish IDVs, Thr was present in both the D/OK and D/660 sequences. At residue 289, Val was present in the Danish D/OK sequences, while Ser was present in the Danish D/660 sequences. Others have found Ala, Ser, or Thr at residue 289, and showed that with the presence of Thr, a new site for N-linked glycosylation was created [41]. Different N-linked glycosylation sites were found in the Danish IDVs, where three of the D/OK sequences differed from the others by lacking the glycosylation site at residue 513, which has also been reported from France [14]. Since experimental studies have yet to be conducted on Danish IDVs, it is unclear if the different glycosylation profiles affect the transmission and/or pathogenicity of these viruses.

This study is the first to describe the detection and full-genome characterization of IDVs in Denmark and in Danish calves. Since this virus has been found in other animal species such as swine, feral swine, camelids, ruminants, and horses in other countries, we are planning future systematic virological and serological screenings to elucidate the presence and diversity of IDVs in Denmark in these species.

## Figures and Tables

**Figure 1 viruses-14-00423-f001:**
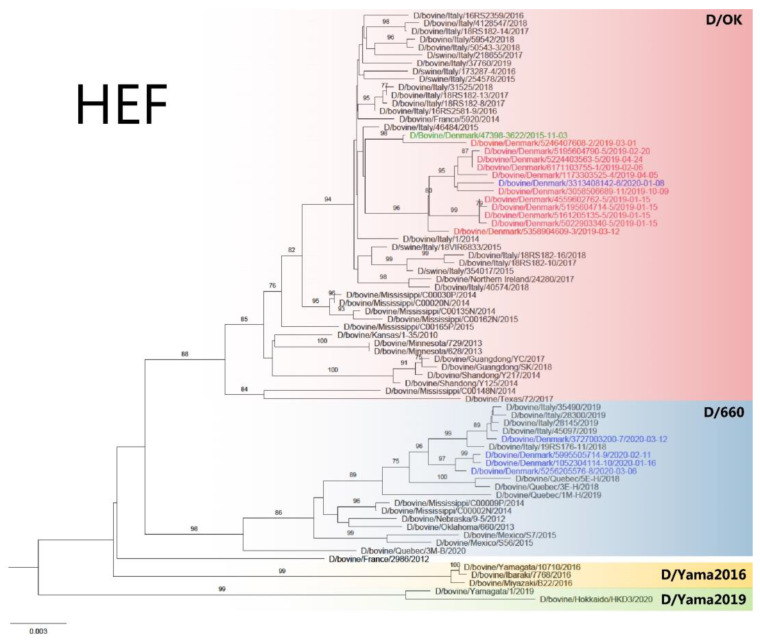
The maximum-likelihood phylogenetic tree of the IDV HEF gene. The highlighted areas denote which lineage the viruses belong to. The sequence of the virus detected in Denmark in 2015 is shown in green text, depicted in red are the IDV viruses found in 2019, and those from 2020 are in blue. All bootstrap values < 70% have been removed. The scale bar indicates nucleotide substitutions per site per year.

**Figure 2 viruses-14-00423-f002:**
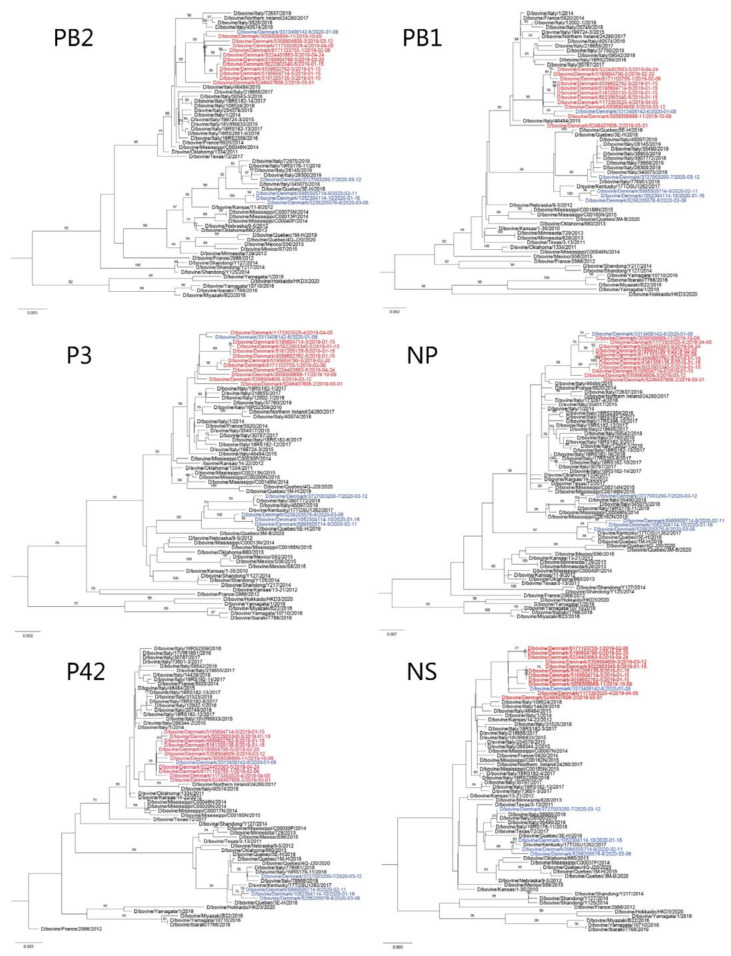
The maximum-likelihood phylogenetic trees of the IDV PB2, PB1, P3, NP, P42, and NS genes. The sequences of IDVs detected in Denmark in 2019 are depicted in red, and depicted in blue are sequences from viruses found in 2020. All bootstrap values < 70% have been removed. The scale bar indicates nucleotide substitutions per site per year.

**Table 1 viruses-14-00423-t001:** IDV-positive samples included in the study. The herd of each strain is identified by a number. The date of sampling, source, and production type are reported together with animal information (sex, age) and clinical registrations (temperature, cough, nasal and eye discharge). The lineage attribution is based on the HEF gene.

Strain	Herd ID	Date of Sampling	Production Type	Source	Sex	Age	Temperature (℃)	Cough *	Nasal Discharge ^■^	Eye Discharge ^■^	Lineage HEF
D/bovine/Denmark/6171103755-1/2019	1	06-02-2019	Dairy	NS	Heifer	3 m	38.5	0	2	1	D/OK
D/bovine/Denmark/5246407608-2/2019	2	01-03-2019	Veal	NS	Bull	2 waa	39.9	0	2	0	D/OK
D/bovine/Denmark/5358904609-3/2019	3	12-03-2019	Veal	NS	Bull	3 m	39.0	0	2	0	D/OK
D/bovine/Denmark/1173303525-4/2019	4	05-04-2019	Veal	NS	Bull	3 m	38.9	0	2	0	D/OK
D/bovine/Denmark/4559602762-5/2019	5	15-01-2019	Veal	NS	Bull	2 waa	38.7	0	2	0	D/OK
D/bovine/Denmark/5022903340-5/2019	5	15-01-2019	Veal	NS	Bull	2 waa	38.9	0	1	1	D/OK
D/bovine/Denmark/5161205135-5/2019	5	15-01-2019	Veal	NS	Bull	2 waa	38.5	0	1	1	D/OK
D/bovine/Denmark/5195604714-5/2019	5	15-01-2019	Veal	NS	Bull	3 m	39.3	0	2	0	D/OK
D/bovine/Denmark/5195604790-5/2019	5	20-02-2019	Veal	NS	Bull	2 waa	39.1	0	2	1	D/OK
D/bovine/Denmark/5224403563-5/2019	5	24-04-2019	Veal	NS	Bull	3 m	38.3	0	1	0	D/OK
D/bovine/Denmark/3313408142-6/2020	6	08-01-2020	Veal	NS	Bull	3 m	39.0	3	2	2	D/OK
D/bovine/Denmark/3727003200-7/2020	7	12-03-2020	Veal	NS	Bull	2 waa	40.5	0	2	2	D/660
D/bovine/Denmark/5256205576-8/2020	8	06-03-2020	Veal	NS	Heifer	3 m	39.6	3	2	2	D/660
D/bovine/Denmark/5995505714-9/2020	9	11-02-2020	Veal	NS	Bull	3 m	39.5	0	2	2	D/660
D/bovine/Denmark/1052304114-10/2020	10	16-01-2020	Veal	NS	Bull	3 m	39.2	2	2	0	D/660
D/bovine/Denmark/3058506689-11/2019	11	09-10-2019	Veal	NS	Bull	3 m	39.7	0	2	0	D/OK
D/bovine/Denmark/4070604622-12/2019	12	16-01-2019	Veal	NS	Bull	3 m	39.5	0	1	1	-
D/bovine/Denmark/47398-3622/2015	13	03-11-2015	Veal	BAL	nd	nd	nd	nd	nd	nd	D/OK

NS: nasal swab, BAL: bronchoalveolar lavage, m: month, waa: weeks after arrival, -: sequencing was not possible due to low viral load. * 0: no coughing observed; 1: a single cough when provoked; 2: repeated coughs when provoked or a single spontaneous cough; 3: repeated unprovoked coughs. ^■^ 0: normal, no signs of discharge; 1: clear, serous discharge, and no sign of mucopurulent or purulent exudate; 2: mucopurulent or purulent discharge.

## Data Availability

The sequences are available at GenBank (www.ncbi.nlm.nih.gov/genbank) with the accession numbers OM468185–OM468297.

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
