# Peer review of "Characterization of Influenza D Virus in Danish Calves"

_viruses, 2022, doi:10.3390/v14020423_

Round 1

Reviewer 1 Report

 “The manuscript is well written, and the introduction presents a sufficient background to the topic. The Material and methods part covers most of the essential issues. The results represent the first influenza D report from Scandinavia. See minor suggestions in the attached file.

My only concern is the need for ethical permission for the sampling if, as the authors indicated in the discussion, “ the samples did thus not originate from diagnostic submissions.” Then as a part of the research project, you would generally need ethical approval before invasive sampling.

Author Response

Dear Reviewer 1

See attached reply.

Thanks

Reviewer 2 Report

Goecke et al screened 100 cattle herds in Denmark for the presence of influenza D virus. Full genome sequence were obtained for almost all positive samples and the phylogenetic relationship of the strains with each other and other publicly available full genomes were assessed for each of the 7 gene segments. The authors also provide some clinical data on IDV positive herd. The investigations are sound and well presented. However, some improvements are suggested. Especially, the scoring system is not clear and warrants clarification. The comparison of clinical data between IDV negative and positive herd is lacking, as well the presence of other pathogens in the herds – data that is most likely available to the authors.

I recommend publication of the manuscript in Viruses when the authors have addressed the comments.

Major comments

Lines 88-89: The introduction is well written and summarizes the state-of-the art on influenza D virus. However, it does not really bring out why the study that was performed was necessary. A small restructuring of the introduction as well as better putting the novelty and interest of this study would be welcome.

Lines 104-105: Please provide a brief summary of the technique used as high-throughput real-time PCR system to avoid the necessity to read through another article to get the essence of the technique.

Lines 104-105: The authors mention that some samples were tested as individual samples, others as pools.

Please describe the criteria for deciding which samples were tested individually or in pools. How many samples were pooled together and how? Was the analytical sensitivity of the pooling assessed?

Lines 92-105: How many animals were tested in total? How many per herd?

Lines 116-124: The scoring system used to assess clinical signs is only partially described in the Material and Method section and as a footnote of Table 1, but not fully at either place. There are also some contradictions between the scoring descriptions.

For instance, the authors wrote (lines 120-121): “For coughing, the highest score (three) was given if at least one spontaneous cough was counted.”. In footnote of Table 1, the authors wrote: “0: No coughing observed, 1: A single cough when provoked, 2: Repeated coughs when provoked or a single spontaneous, 3: Repeated unprovoked coughs.” It is therefore not clear which scores was given to a calf with one spontaneous cough, 2 or 3?

It is also not clear if a rectal temperature <39.3 was scored as 0 and >39.3 as 1?

The objective of a clinical scoring system is often to integrate the contribution of multiple symptoms to provide an overall disease severity assessment. How were the scores for each clinical parameter integrated together?

The scoring system would therefore gain in significance if the criteria and scores would be described in a clearer and systematic way, with all scores for each sign listed explicitly.

Lines 205-215 and Table 1: Only the clinical symptoms of IDV positive herds are listed in a descriptive manner. The scores of IDV positive herds are not compared to IDV negative herds. Therefore, the potential clinical impact of IDV on the animals cannot be estimated here. The lack of comparison undermines the use of a clinical scoring, which would be valuable.

Lines 305: How many positive animals? What is the overall prevalence of IDV observed during the study?

Overall, it is not obvious if the study is a subsequent study on the same respiratory samples used for Goecke et al. Front Vet Sci. 2021 Jun 24;8:677993 or not. If it is the case, it should be made very clear.

If it is a subsequent study, then the data to assess whether the clinical signs observed in IDV positive herds might be due to (co-)infection by other pathogens is available to the authors (lines 308-309) and should be presented.

What is the bovine import situation in Denmark? The authors mention that Denmark imports very limited number of cattle head, but is the main origin of those animals known? Could trade with European countries such as France and Italy possibly explain the introduction of a second IDV lineage in Denmark as observed in 2020? Does Denmark export live cattle to European countries?

Minor comments:

Line 15: The authors wrote “RNA was extracted from the positive samples…”. This sentence is misleading since RNA extraction is a necessary step prior to detection of IDV positive samples by the high-throughput real-time PCR platform. Please rephrase.

Lines 108: The authors wrote “The most positive sample… “. Please rephrase in a more scientific way, for instance “samples with the lowest Ct value” or “samples with the highest viral load”.

Lines 227-228: The accession numbers of the sequences obtained in this study would be better located in the Material and Method section.

Figures: Low boostrap values are not very informative and should be removed from the corresponding notes in the phylogenetic trees. Very often, only bootstrap values >70% are reported.

Author Response

Dear Reviewer 2

See attached reply.

Thanks
